# Cardiac Rehabilitation Increases Plasma Klotho Levels

**DOI:** 10.3390/jcm13061664

**Published:** 2024-03-14

**Authors:** Ana María Pello Lázaro, Koldo Villelabeitia Jaureguizar, Juan Antonio Franco Peláez, Ana Venegas-Rodriguez, Álvaro Aceña, Andrea Kallmeyer, Ester Cánovas, María Luisa González-Casaus, Nieves Tarín, Carmen Cristóbal, Carlos Gutiérrez-Landaluce, Ana Huelmos, Óscar González-Lorenzo, Joaquín Alonso, Lorenzo López-Bescós, Jesús Egido, Ignacio Mahillo-Fernández, Jairo Lumpuy-Castillo, Óscar Lorenzo, José Tuñón

**Affiliations:** 1Department of Cardiology, IIS-Fundación Jiménez Díaz, 28040 Madrid, Spainaacena@fjd.es (Á.A.); ester.canovas@fjd.es (E.C.); ogonzalez@quironsalud.es (Ó.G.-L.); jtunon@quironsalud.es (J.T.); 2Faculty of Medicine, Autónoma University, 28029 Madrid, Spain; jegido@quironsalud.es (J.E.); olorenzo@fjd.es (Ó.L.); 3Department of Rehabilitation, Hospital Universitario Infanta Elena, 28340 Madrid, Spain; 4Unitof Mineral Metabolism, Department of Laboratory Medicine, Hospital La Paz, 28046 Madrid, Spain; mlgcasaus@gmail.com; 5Department of Cardiology, Hospital Universitario de Móstoles, 28935 Madrid, Spain; 6Faculty of Medicine, Rey Juan Carlos University, Alcorcón, 28922 Madrid, Spainllopez@fhalcorcon.es (L.L.-B.); 7Department of Cardiology, Hospital Universitario de Fuenlabrada, 28942 Madrid, Spain; 8Department of Cardiology, Hospital Universitario Fundación Alcorcón, 28922 Madrid, Spain; 9Department of Cardiology, Hospital de Getafe, 28905 Madrid, Spain; 10CIBERDEM (Diabetes and Associated Metabolic Diseases Networking Biomedical Research Centre), 28029 Madrid, Spain; jairo.lumpuy@quironsalud.es; 11Renal, Vascular and Diabetes Research Laboratory, IIS-Fundación Jiménez Díaz, 28040 Madrid, Spain; 12Laboratory of Vascular Pathology, IIS-Fundación Jiménez Díaz, 28040 Madrid, Spain; 13Biostatistics and Epidemiology Unit, IIS-Fundación Jiménez Díaz, 28040 Madrid, Spain; imahillo@fjd.es; 14Centro de Investigación Biomédica en Red Enfermedades Cardiovaculares (CIBERCV), ISCIII, 28029 Madrid, Spain

**Keywords:** fibroblast growth factor-23, klotho, cardiovascular risk, acute coronary syndrome, mineral metabolism biomarkers

## Abstract

**Background**: Mineral metabolism (MM), mainly fibroblast growth factor-23 (FGF-23) and klotho, has been linked to cardiovascular (CV) diseases. Cardiac rehabilitation (CR) has been demonstrated to reduce CV events, although its potential relationship with changes in MM is unknown. **Methods**: We performed a prospective, observational, case-control study, with acute coronary syndrome (ACS) patients who underwent CR and control patients (matched by age, gender, left ventricular ejection fraction, diabetes, and coronary artery bypass grafting), who did not. The inclusion dates were from August 2013 to November 2017 in CR group and from July 2006 to June 2014 in control group. Clinical, biochemical, and MM biomarkers were collected at discharge and six months later. Our objective was to evaluate differences in the modification pattern of MM in both groups. **Results**: We included 58 CR patients and 116 controls. The control group showed a higher prevalence of hypertension (50.9% vs. 34.5%), ST-elevated myocardial infarction (59.5% vs. 29.3%), and treatment with angiotensin-converting enzyme inhibitors (100% vs. 69%). P2Y12 inhibitors and beta-blockers were more frequently prescribed in the CR group (83.6% vs. 96.6% and 82.8% vs. 94.8%, respectively). After six months, klotho levels increased in CR patients whereas they were reduced in controls (+63 vs. −49 pg/mL; *p* < 0.001). FGF-23 was unchanged in the CR group and reduced in controls (+0.2 vs. −17.3 RU/dL; *p* < 0.003). After multivariate analysis, only the change in klotho levels was significantly different between groups (+124 pg/mL favoring CR group; IC 95% [+44 to +205]; *p* = 0.003). **Conclusions**: In our study, CR after ACS increases plasma klotho levels without significant changes in other components of MM. Further studies are needed to clarify whether this effect has a causal role in the clinical benefit of CR.

## 1. Introduction

After an acute coronary syndrome (ACS), completion of a cardiac rehabilitation (CR) program is recommended in clinical guidelines, in order to reduce cardiovascular (CV) and all-cause mortality, CV hospitalizations, and myocardial infarctions [1,2,3]. CR programs provide a set of multidisciplinary interventions, including physical exercise, lifestyle modification, nutritional counselling, CV risk factors adjustment, psychological support and drug therapy optimization, with the aim of improving outcomes and promoting secondary prevention in these patients [2,3].

Mineral metabolism involves a complex metabolic pathway that has been associated with the pathogenesis of CV diseases [4]. Although the most widely recognized components of this pathway are vitamin D, phosphate, and parathyroid hormone, other molecules such as fibroblast growth factor (FGF)-23 and its co-receptor klotho have been linked to cardiotoxic effects [5]. Higher levels of FGF-23 have been associated with congestive heart failure, left ventricular hypertrophy, hypertension and atrial fibrillation [6,7,8,9], and they likely have prognostic value for predicting CV death and all-cause mortality [10,11]. These effects have been demonstrated in patients with chronic kidney disease (CKD) and normal kidney function [12,13,14]. On the other hand, plasma klotho is independently associated with cardiovascular disease in adults and a cardioprotective role of klotho has already been described. Klotho is involved in vascular and endothelial homeostasis, where it prevents vascular calcification, exerts pleiotropic effects and plays a key role in regulating nitric oxide availability in the endothelium through suppression of oxidative stress and inflammatory responses [15,16,17,18,19,20,21]. In addition, circulating klotho maintains endothelial integrity and protects against vascular permeability, preventing medial hypertrophy and perivascular fibrosis [21].Consequently, disruption of klotho expression is involved in atherosclerosis as well as in other multiple aging phenotypes. Lower levels of klotho are present in patients with higher CV risk [15], and the opposite is true even after adjusting for other CV risk factors [22].

Although exercise is known to decrease plasma levels of some pro-inflammatory molecules [23], there is a dearth of data concerning the relationship between exercise and FGF-23 and klotho. It is known that CR can improve CV fitness and CV risk factors, reduce inflammation and oxidative stress, and enhance endothelial function, but whether these effects could contribute to regulation of FGF-23 and klotho remains unclear. For that reason, we designed this study with the main objective of exploring the possible impact of CR on MM biomarkers in patients after an ACS.

## 2. Materials and Methods

### 2.1. Patients

This is a sub-study of BACS & BAMI (Biomarkers in Acute Coronary Syndrome & Biomarkers in Acute Myocardial Infarction) and BIOMACIK (benefits of physical training on functional capacity and plasma biomarkers) studies.

The BACS & BAMI study included patients admitted to five hospitals in Madrid who had ACS (either non-ST-elevated acute coronary syndrome (NSTEACS) or ST-elevated acute myocardial infarction (STEMI)). NSTEACS was defined as angina at rest lasting >20 min in the previous 24 h, or new-onset class III to IV angina, along with transient ST depression or T wave inversion on the electrocardiogram considered diagnostic by the attending cardiologist, and/or troponin elevation. STEMI was defined as symptoms compatible with angina lasting >20 min, ST elevation in 2 adjacent leads on the electrocardiogram without response to nitroglycerin and troponin elevation. Exclusion criteria were age >85 years, coexistence of other significant cardiac disorders except left ventricular hypertrophy secondary to hypertension, coexistence of any illness or toxic habits that could limit patient survival, impossibility to perform revascularization when indicated and subjects in whom follow-up was not possible [24,25]. Between July 2006 and June 2014, 964 patients met inclusion criteria prospectively, and completed a follow-up visit six months after the ACS. None of these patients was included in a CR program due to lack of availability at the hospitals.

The BIOMACIK study included patients admitted to Infanta Elena University Hospital in Valdemoro (Madrid, Spain) with NSTEACS or STEMI and referred to a CR program at discharge. Inclusion and exclusion criteria were identical to those of BACS & BAMI, along with other exclusion criteria: residual ischemia pending further revascularization, mental disability which prevented proper adhesion to the program, presence of moderate to severe chronic obstructive pulmonary disease, and physical limitations that precluded the performance of protocolized exercise. Between August 2013 and November 2017, 59 patients were prospectively included. One of them did not undergo blood sample extraction and was excluded, leaving 58 patients for further analysis who were followed up for six months after ACS.

The research protocol of the two studies was approved by the ethics committees of the participating hospitals, and all patients signed informed consent documents.

### 2.2. Study Design

The aim of our study was to explore the impact of the CR program on MM biomarkers among patients with ACS. We designed a prospective, observational case-control study with BIOMACIK study patients who had completed CR in the months following ACS and BACS & BAMI study patients who had not. Thus, 58 patients coming from the former group (rehabilitation group) and 116 patients from the later (control group) (1:2 ratio) were matched by the following characteristics: age, gender, diabetes, left ventricular systolic dysfunction, and previous coronary artery bypass grafting.

Patients undergoing CR were enrolled in an intervention protocol that was individualized to everyone. The mode of exercise training was a cycle ergometer with 40 min per session, 3 days per week (total of 24 sessions over 2 months). They were also taught to devise a home walking program for the days on which they did not have to attend sessions in the hospital.

In both BACS & BAMI and BIOMACIK, a complete data set of clinical variables was collected on admission (visit 0) and at six months later (visit 1). Similarly, 12-h venous blood extractions were collected in ethylene diamine tetraacetate (EDTA) tubes at the same time.

### 2.3. Biomarker and Analytical Studies

Venous blood samples were centrifuged at 2500× *g* for 10 min. Plasma was stored at −80 °C in the biobank of Fundación Jiménez Diaz. Plasma determinations were performed at the MM laboratory of Hospital La Paz (Madrid, Spain) and the Vascular and Biochemistry laboratories of Fundación Jiménez Díaz. The investigators who performed the laboratory studies were unaware of clinical data. Plasma calcidiol levels were quantified by chemiluminescent immunoassay (CLIA) on the LIAISON XL analyzer (LIAISON 25OH-vitamin D TOTAL assay, DiaSorin, Saluggia, Italy), FGF-23 was measured by an enzyme-linked immunosorbent assay which recognizes epitopes within the carboxyl-terminal portion of FGF-23 (human FGF-23, C-Term, Immutopics Inc., San Clemente, CA, USA), klotho levels were gauged using ELISA (human soluble alpha klotho assay kit, Immuno-Biological Laboratories Co., Hokkaido, Japan), intact PTH was analyzed by a second-generation automated chemiluminescent method (Elecsys 2010 platform, Roche Diagnostics, Mannheim, Germany), and phosphate was determined by an enzymatic method (Integra 400 analyzer, Roche Diagnostics, Mannheim, Germany). N-terminal pro-brain natriuretic peptide (NT-proBNP) levels were assessed by immunoassay (VITROS, Ortho Clinical Diagnostics, Raritan, NJ, USA), and high-sensitivity C-reactive protein (hs-CRP) by latex-enhanced immunoturbidimetry (ADVIA 2400 Chemistry System, Siemens, Munich, Germany). Lipids, glucose, hemoglobin, glycosylated hemoglobin, proteins, and creatinine determinations were performed by standard methods (ADVIA 2400 Chemistry System, Siemens, Munich, Germany). The eGFR was calculated using the CKD epidemiology collaboration equation [26].

### 2.4. Statistical Analysis

Quantitative variables are presented as medians and interquartile ranges, whereas qualitative are shown as numbers and percentages. Comparisons between quantitative variables were performed by using a Student’s t-test in those which met the normality assumption and a Mann–Whitney test for those which did not. Analysis of normality was performed with the Kolmogorov–Smirnov test, where *p* < 0.05 meant that the variable did not meet the normality assumption. Qualitative variables were compared with the Chi-square Test (or with Fisher’s exact test when appropriate).

All analytical biomarkers were analyzed either at visit 0 or visit 1. The change of every variable was calculated as the subtraction of the visit 1 value minus visit 0, and results are displayed as means and standard deviations. Comparisons between groups were initially made in a univariate fashion with the same method described above for quantitative variables. Due to the non-randomized nature of our study, we decided to carry out a multivariate analysis as well, including baseline characteristics that were significantly different between groups at visit 0. Thus, when the change of an analytical variable met the normality assumption, we performed a multivariate lineal regression analysis, whereas a multivariate quantile regression analysis was performed when the variable did not meet the assumption. Results are presented as means (or medians in the latter) difference in change between groups and its 95% confidence interval (CI).

Analyses were performed with IBM SPSS Statistics for Windows Version 19.0 (IBM Corp., Armonk, NY, USA), and were considered significant when the “*p*” value was lower than 0.05 (two-tailed).

## 3. Results

### 3.1. Baseline Characteristics

Our study included 174 patients: 58 in the CR group and 116 in the control group. Patients were predominantly male and the median age was 56 years (Table 1). Hypertension and STEMI as the index event were more frequent in the control group, and the number of coronary vessels affected and the percentage of patients receiving percutaneous coronary intervention (PCI) were higher in the CR group. The remaining variables were balanced between groups and the percentage of patients fully revascularized was similar.

Regarding pharmacological treatment at discharge, angiotensin-converting enzyme inhibitors (ACEI) were more frequently used in the control group, whereas P2Y12 inhibitors and beta-blockers were used more often in the CR group (Figure 1). Baseline creatinine, and high- and low-density plasma lipoprotein (HDL and LDL) levels were lower in the CR group, and there were no differences in other biomarkers, including the components of mineral metabolism (Figure 2).

### 3.2. Modulation of Biomarkers after Six Months

The next step was to perform a univariate comparison of the magnitude of the change of every parameter between visit 1 and visit 0, as shown in Table 2. It can be observed that only two parameters reached significant statistical differences between groups: FGF-23 (−17.3 vs. +0.2 RU/dL; *p* = 0.003) and klotho (−49 vs. +63 pg/mL; *p* < 0.001). There were no significant changes in difference of other MM components.

In order to avoid confounding factors, we performed a new comparison of the magnitude of the change between groups by adjusting for the following variables, which had significant differences at visit 0: hypertension, type of ACS (STEMI or NSTEACS) and prescription of ACEI, beta-blockers, and P2Y12 inhibitors at discharge (Table 3). Thus, patients in the CR group showed a higher increase in klotho (+124 pg/mL, CI 95%: +44 to +205; *p* = 0.003) and LDL levels (+20 mg/dL; CI 95%: +6 to +35; *p* = 0.007). There were no other significant differences between groups.

## 4. Discussion

The main finding of our study is that a CR program is associated with an increase of klotho levels in ACS patients.

### 4.1. Characteristics of the Population

Patients were predominantly male, which is in line with the higher prevalence of CAD in males described in previous literature [27]. The prevalence of CV risk factors in our population was high, as expected for patients who have experienced an ACS [1]. Overall, clinical baseline characteristics were well-balanced, except for an increased presence of hypertension and STEMI as the index event in the control group, more disease vessels and a higher percentage of PCI in CR patients. Although the use of ACEI at discharge was higher in the control group, probably driven by the higher prevalence of hypertension, the use of P2Y12 inhibitors and beta-blockers was more frequent in the CR group.

Some of these differences may be due to an earlier recruitment of the control group as compared with CR patients. In fact, an increase in NSTEMI patients as compared to STEMI in the recent decades has been reported primarily due to modifications and improvements in the diagnosis of NSTEMI [28]. Data from the French registry of STEMI or NSTEMI highlight a significant rise not only in the utilization of early angiography, but also in the use of PCI during the hospital stay [28,29].

### 4.2. Adjusted Comparison of Changes in Biomarkers between Groups

In our study, patients of the CR group showed a higher increase in klotho levels, without differences in the other MM biomarkers. Klotho is a transmembrane protein synthesized in the kidney that forms complexes with FGF-23. There are two isoforms of membrane klotho (α and β-klotho) which may be shed by secretases to release a soluble form (S-klotho) into the blood [4]. This is the one we are referring to in this study, and we measured it in plasma. Data previously published confirm that lower plasma S-klotho levels are associated with several CV risk factors (higher body mass index, smoking, alcohol consumption, triglycerides, and total cholesterol), with higher cardiometabolic risk scores, especially in adult patients, and with higher probability of CV disease, even after adjusting for traditional CV risk factors [22,30,31].

In our study, the increase in klotho levels is greater in the CR group, a finding consistent with some previously published data on the relationship between exercise and S-klotho regulation. Physical activity is known to play a favorable role in atherosclerosis development and prevents some causes of morbidity and mortality, especially those associated with aging [23,32]. There is some evidence of the relation between exercise and S-klotho regulation. This effect could partially explain the anti-atherogenic effects of exercise. Thus, in the study of Mostafidi et al., klotho levels (measured the morning after their last evening exercise training) were significantly higher in trained athletes than in healthy subjects [33]. Another small study of healthy adults who performed a single type of high-intensity standardized exercise concludes that exercise is associated with increased klotho levels immediately after exercise, with a slight decline in phosphate levels [34]. A meta-analysis of 12 reports involving 621 participants with different health conditions (healthy subjects, patients with CAD, CKD, and on hemodialysis) showed that chronic exercise training for a minimum of 12 weeks significantly increased klotho concentration regardless of the individual’s health condition or the specific exercise intervention [35]. Additionally, the duration and volume of the exercise protocol appeared to influence S-klotho concentration. A possible inverted U-shaped curve was observed, indicating a dose-response relationship between klotho changes and the volume of the training protocol, with approximately 150 min per week showing the highest magnitude of klotho change [35]. The increase in klotho levels following exercise could partially explain its anti-atherogenic effects.

As far as we know, our study is the first to demonstrate that an established CR program in patients with a recent ACS is associated with an increase of plasma klotho levels even after extensive adjustment for confounding factors. Nevertheless, a recent study examined the relationship between plasma klotho levels and the clinical condition of patients with chronic obstructive pulmonary disease who participated in a three-week pulmonary rehabilitation program. The study concluded that klotho levels did not change significantly after pulmonary rehabilitation and did not seem to correlate with certain clinical parameters (respiratory function, walking distance, or grip strength, among others) [36]. This contradiction may be attributed to two factors. Firstly, a different patient profile, as these were patients with respiratory disorders. Secondly, the pulmonary rehabilitation program was more focused on daily respiratory muscle training, controlled breathing techniques, and chest-wall mobilization than on bicycle training.

FGF-23 is a potent negative regulator of αKlotho expression, although the exact mechanism of this regulation is not yet known. For this reason, one would expect to find inverse behavior of these two molecules in our patient population, and yet we have not found this. We performed a correlation analysis between klotho and FGF-23 plasma levels, either at visit 0 or visit 1. The results did not show a significant correlation (visit 0 r = −0.18; *p* = 0.02 and visit 1 r = −0.064; *p* = 0.402).

Evidence regarding the behavior of FGF-23 with exercise is scant, limited to CKD patients and, to some extent, controversial [37]. In one study involving patients on hemodialysis, dynamic resistance exercise training was found to increase klotho and reduce FGF-23 and PTH levels [38]. Another study involving patients with stage 2 CKD found that six months of resistance exercise increased klotho levels and reduced FGF-23 [39]. However, in a sub-study of the RENEXC (randomized controlled trial in CKD) involving patients with stage 3–5 CKD, 12 months of strength or balance training and aerobic exercise had no effect on FGF-23 levels [40].

A recent study of 172 patients with heart failure and preserved ejection fraction showed that higher FGF-23 levels were independently linked to lower peak oxygen consumption and shorter six-minute walk distance at the initial assessment. However, these associations did not remain significant over the 24-week study period, which suggests a possible time-dependent effect or other contributing factors [41]. In this regard, we have found that elevated FGF-23 levels were independently associated with an increased risk of composite CV death or hospitalization for heart failure after ACS [25]. These findings are consistent with other published data [42]. The association of higher FGF-23 levels with the development of heart failure has also been described in patients with acute heart failure and even in the general population [43,44].

The results of our study reveal a higher increase in LDL levels in the CR group, which is a striking finding that is not easily explained. The most robust evidence suggests that CR can effectively reduce all the components of the lipid profile while improving HDL-C levels [45,46]. This improvement in lipid profile parameters appears to be independent of the administered lipid-lowering treatment and of the control of other risk factors during the CR period [47]. In past research on the combined effects of diet and exercise on lipoprotein-cholesterol levels, the most common finding has been an increase in HDL-C of approximately 5%. However, changes in LDL-C are more heterogeneous due to the differential effects of dietary cholesterol, fat, carbohydrate, and physical-activity intensity on LDL-C levels and the composition of the LDL particles. For example, vigorous activity favors a switch toward larger LDL particles [48,49,50]. The fact that the variation of LDL levels with exercise is more heterogeneous could be the reason for the increase in LDL in the CR group in our study, where the physical exercise program of each patient was individualized.

## 5. Limitations

The main limitation of our study is that it is an observational, case-control study with two populations that were recruited at different time points. Despite the statistical adjustments made to draw conclusions, this should be considered a significant limitation when extrapolating the data to other populations and with great caution in the conclusions from this work. The second limitation is that the number of patients is small, especially in the CR group. Finally, most patients had a low-risk profile, with a small percentage having ventricular dysfunction, previous heart failure, or comorbidities. These characteristics could limit the generalizability of our results to populations with a higher cardiovascular risk.

## 6. Conclusions

CR plays a crucial role in secondary prevention improving the quality of life and long-term prognosis among individuals with cardiovascular disease. After an ACS, CR is associated with increased plasma klotho levels, which may be one of the beneficial mechanisms of CR. In this regard, FGF-23 levels and other MM biomarkers did not change subsequent to CR, even after adjusting analysis. These findings reflect the complexity of mineral metabolism with different mechanisms of action of the components although with a common pathophysiological pathway. Further research is needed to determine if the increase in klotho levels among patients undergoing a CR program after an ACS results in improved functional capacity or plays a beneficial pathophysiological role in the atherosclerosis process with improvement of the patient’s prognosis.

## Figures and Tables

**Figure 1 jcm-13-01664-f001:**
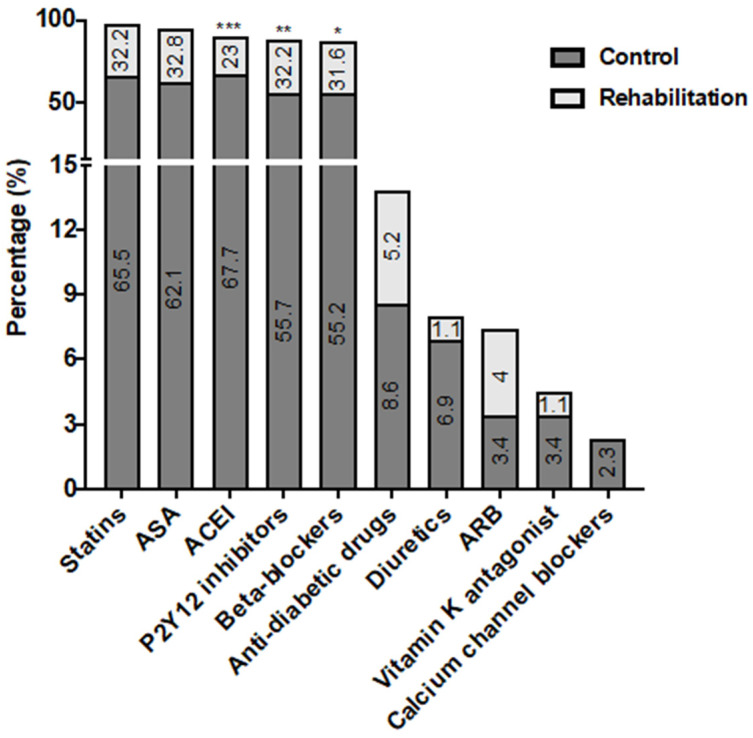
Pharmacological treatments in patients with acute coronary syndrome. In V0, pharmacological treatments were recorded in the control (n = 116) and rehabilitation (n = 58) groups. Significant differences were found between both groups (* *p* < 0.05, ** *p* < 0.01, *** *p* < 0.001, vs. control). ASA: Acetylsalicylic acid. ACEI: Angiotensin-converting enzyme inhibitors. ARB: Angiotensin receptor blockers.

**Figure 2 jcm-13-01664-f002:**
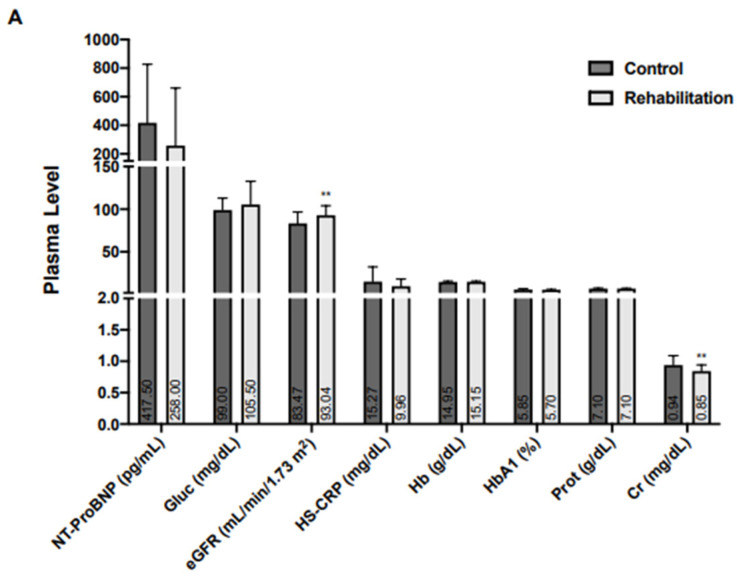
Baseline plasma markers in patients with acute coronary syndrome.Biochemical biomarkers (**A**), mineral metabolism biomarkers (**B**), and the lipid profile (**C**) were recorded in the control (n = 116) and rehabilitation (n = 58) groups. Quantitative variables are presented as median (interquartile range). The level of Cr, HDL-C, and LDL-C were lower in the rehabilitation group. * *p* < 0.05 and ** *p* < 0.01, vs. control. NT-ProBNP, N-terminal pro B-type natriuretic peptide; Gluc, Glucose; eGFR, Estimated glomerular filtration rate; HS-CRP High-sensitivity C-reactive protein; Hb, Hemoglobin; HbA1c, Glycosylated hemoglobin; Prot: Proteins, Cr: Creatinine; FGF-23, Fibroblast growth factor-23; PTH, Parathormone, TG, Triglycerides; TC, Total cholesterol; LDL-C, Low-density lipoprotein-cholesterol; HDL-C, High-density lipoprotein-cholesterol; Lp(a), Lipoprotein (a).

**Table 1 jcm-13-01664-t001:** Baseline clinical characteristics.

	Overall (N = 174)	Control Group (N = 116)	Rehabilitation Group (N = 58)	*p*
Age (years), median (IQR)	56 (47–63)	56 (47–63)	56 (47–63)	0.969
Male gender, n (%)	138 (79.3)	92 (79.3)	46 (79.3)	1
Hypertension, n (%)	79 (45.4)	59 (50.9)	20 (34.5)	0.039
Tobacco consumption, n (%):				0.185
Former	51 (29.3)	39 (33.6)	12 (20.7)	
Active	94 (54.0)	58 (50.0)	36 (62.1)	
Never	29 (16.7)	19 (16.4)	10 (17.2)	
Diabetes, n (%)	27 (15.5)	18 (15.5)	9 (15.5)	1
Hypercholesterolemia, n (%)	113 (64.9)	74 (63.8)	39 (67.2)	0.652
BMI (kg/m^2^), median (IQR)	27.6 (25.3–30.5)	27.5 (25.3–29.9)	27.8 (25.1–32.6)	0.84
Previous heart failure, n (%)	0 (0)	0 (0)	0 (0)	-
Atrial fibrillation, n (%)	2 (1.1)	1 (0.9)	1 (1.7)	1
Ischemic heart disease, n (%)	19 (10.9)	15 (12.9)	4 (6.9)	0.213
LVEF < 40%, n (%)	7 (4.0)	5 (4.3)	2 (3.4)	1
Previous stroke, n (%)	2 (1.1)	1 (0.9)	1 (1.7)	1
COPD or SAHS, n (%)	15 (8.6)	8 (6.9)	7 (12.1)	0.263
Peripheral vascular disease, n (%)	4 (2.3)	4 (3.4)	0	0.303
Functional class I, n (%)	155 (92.8)	103 (90.4)	52 (98.1)	0.106
Type of index event, n (%):				<0.001
- NSTEMI	88 (50.6)	47 (40.5)	41 (70.7)	
- STEMI	86 (49.4)	47 (40.5)	17 (29.3)	
Full revascularization, n (%)	127 (73.0)	90 (77.6)	37 (63.8)	0.056
Number of vessels involved, n (%)				0.036
- 0	14 (8.0)	12 (10.3)	2 (3.4)	
- 1	103 (59.2)	70 (60.3)	33 (56.9)	
- 2	40 (23.0)	26 (22.4)	14 (24.1)	
- 3	17 (9.8)	8 (6.9)	9 (15.5)	
Type of revascularization, n (%)				0.028
- None	20 (11.5)	18 (15.5)	2 (3.4)	
- Percutaneous	147 (84.5)	92 (79.3)	55 (94.8)	
- CABG	7 (4.0)	6 (5.2%)	1 (1.7)	

BMI: Bodymass index. CABG: Coronary artery bypass graft. COPD: Chronic obstructive pulmonary disease. IQR: Interquartile range. LVEF: Left ventricular ejection fraction. SAHS: Sleep apnea-hypopnea syndrome. STEMI: ST-elevated myocardial infarction.

**Table 2 jcm-13-01664-t002:** Changes in analytical parameters from baseline to visit 1.

	Control Group (N = 116)	Rehabilitation Group (N = 58)	*p* *
**Biochemical Biomarkers (V1-V0):**			
Glucose (mg/dL)	−3.4 (27.3)	−7.3 (43.51)	0.249
Creatinine (mg/dL)	−0.02 (0.32)	−0.01 (0.13)	0.911
eGFR (mL/min/1.73 m^2^)	+1.5 (21.9)	+0.5 (10.0)	0.681
Glycosylated hemoglobin (%)	−0.2 (0.5)	−0.1 (0.7)	0.320
Proteins (g/dL)	−0.3 (0.8)	−0.3 (1.0)	0.284
NT-ProBNP (pg/mL)	−442 (903)	−422 (822)	0.725
HS-CRP (mg/dL)	−19.9 (22.9)	−14.7 (20.7)	0.076
**MM Biomarkers:**			
Phosphate (mg/dL)	−0.2 (0.7)	−0.1 (0.7)	0.306
Calcidiol (ng/dL)	+2.3 (11.9)	+3.7 (12.5)	0.482
Parathormone (pg/dL)	+9.8 (20.3)	+10.6 (23.3)	0.477
FGF-23 (RU/dL)	−17.3 (113.5)	+0.2 (86.8)	0.003
Klotho (pg/mL)	−49 (223)	+63 (129)	<0.001
**Lipid Profile:**			
Triglycerides (mg/dL)	−37 (81)	−22 (61)	0.177
HDL cholesterol (mg/dL)	+0.2 (11.0)	+2.3 (7.7)	0.152
Non-HDL cholesterol (mg/dL)	−49 (40)	−54 (49)	0.494
LDL cholesterol (mg/dL)	−42 (37)	−34 (37)	0.197
Total cholesterol (mg/dL)	−49 (42)	−51 (52)	0.732
Lipoprotein(a) (mg/dL)	−2.0 (14.9)	−2.5 (15.4)	0.210

eGFR: Estimated glomerular filtration rate. FGF-23: Fibroblast growth factor-23. HDL: High-density lipoprotein. HS-CRP: High-sensitivity C-reactive protein. MM: metabolism mineral. LDL: Low-density lipoprotein. NT-ProBNP: N-terminal pro B-type natriuretic peptide. V0: at admission. V1: 6 months later. Data are shown as mean differences between visit 1 and visit 0 and standard deviation.* *p* was performed with univariate comparison.

**Table 3 jcm-13-01664-t003:** Adjusted comparison of changes in analytical variables between groups.

	Estimated Adjusted Difference between Groups *	95% Confidence Interval	*p*
**Biochemical Biomarkers:**			
Glucose (mg/dL) **	+4	−4 to +12	0.342
Creatinine (mg/dL) **	+0.02	−0.08 to +0.12	0.694
eGFR (mL/min/1.73 m^2^)	−0.5	−8.1 to +7.1	0.892
Glycosylated hemoglobin (%) **	+0.3	0.0 to +0.6	0.077
Proteins (g/dL) **	+0.2	−0.1 to 0.5	0.132
NT-ProBNP (pg/mL) **	−2	−151 to 148	0.980
HS-CRP (mg/dL) **	−0.2	−7.3 to +7.0	0.967
**MM Biomarkers:**			
Phosphate (mg/dL)	+0.3	0.0 to +0.6	0.055
Calcidiol (ng/dL)	+2.7	−2.2 to +7.5	0.278
Parathormone (pg/dL) **	+3.9	−4.3 to +12.0	0.350
FGF-23 (RU/dL) **	+12.6	−6.9 to +32.1	0.203
Klotho (pg/mL)	+124	+44 to +205	0.003
**Lipid Profile:**			
Triglycerides (mg/dL)	+21	−10 to +51	0.183
HDL cholesterol (mg/dL)	+2	−2 to +6	0.381
Non-HDL cholesterol (mg/dL)	+7	−11 to +25	0.435
LDL cholesterol (mg/dL)	+20	+6 to +35	0.007
Total cholesterol (mg/dL)	+9	−10 to +27	0.359
Lipoprotein(a) (mg/dL) **	−2.0	−5.2 to +1.2	0.216

* Change in rehabilitation group minus change in control group. A positive value denotes a higher increase in the former and a negative value in the latter. Comparisons were performed with multivariate lineal regression for those variables which met the normality assumption (presented as mean difference and 95% confidence interval) and with multivariate quantile regression for those which did not (denoted with ** and presented as median and 95% confidence interval). Adjusting variables were hypertension, type of acute coronary event and use of P2Y12 inhibitors, angiotensin-converting enzyme inhibitors, beta-blockers, and high-intensity statins. eGFR: Estimated glomerular filtration rate. FGF-23: Fibroblast growth factor-23. HDL: High-density lipoprotein. HS-CRP: High-sensitivity C-reactive protein. MM: metabolism mineral. LDL: Low-density lipoprotein. NT-ProBNP: N-terminal pro B-type natriuretic peptide.

## Data Availability

The datasets used and analysed during the current study are available from the corresponding author on reasonable request.

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
