# Peer review of "Cardiac Rehabilitation Increases Plasma Klotho Levels"

_jcm, 2024, doi:10.3390/jcm13061664_

Round 1

Reviewer 1 Report

Comments and Suggestions for Authors

Introduction - I suggest the authors to extend the section by presenting additional data on klotho in the diagnosis and management of patients with cardiovascular pathologies.

It is also necessary to expand the paragraph on the purpose of the present research. 

Materials and method - well systematised section. 

Results - well structured, I suggest the introduction of images based on the statistical data obtained.

Discussion - presents the results of the research in relation to data previously published in the literature.

Conclusions - suggest extending the section

Title - too long, suggest finding a shorter one that will attract the reader.

Reviewer 2 Report

Comments and Suggestions for Authors

Congrats to choose this important part of cardiac patient's. The notes are to improve the quality of work. 

Title is repeat.

In the introduction, physical exercise is not discussed as a component of cardiac rehabilitation programs, despite being described as a facilitating/analytical modulating factor.

The groups have a difference of (1:2) causing doubt in some of the results presented, it would be better to analyze the possibility of a 1:1 analysis.

Reviewer 3 Report

Comments and Suggestions for Authors

Dear Authors,

I would like to state that the study is interesting and I read it with interest. But I would also like to point out that I am conflicted in many places in the article.

1. Abstract: It should be stated whether the study is a retrospective or prospective study, along with its start and end dates.

2. Introduction: It is written very briefly, transitions between topics are disjointed. There were transitions from topic to topic. It should be reorganized and expanded to ensure integrity. Additionally, more coverage should be given about Klotho.

3. Material-Method: It should be noted that it is difficult to understand whether the study is prospective or retrospective. The work seems to have an adventure that started in 2006. So how were FGF-23 and Klotho viewed? Are these two parameters routine in all your admitted patients? It is clear from your centrifuge technique that it is not routine. How it was obtained should be explained? The acceptance and rejection criteria of the study should be explained. Additionally, the reasons for excluding patients who do not meet the rejection criteria should be explained. Were the co-morbid conditions of the patients included in the study determined?

4. Statistics: To show the strength of the study, specificity-sensitivity situations can be made by correlation and ROC curve analysis with FGF-23 and Klotho.

5. Discussion: The authors provided information about FGF-23 throughout the discussion. However, since the goal is to highlight Klotho activity, the discussion should be organized in a way that highlights Klotho.

Kind regards.

Reviewer 4 Report

Comments and Suggestions for Authors

I would like to congratulate authors for the idea. But the study is not properly done .

1) The 2 groups compared are of different times and different studies. There is no uniformity between 2 groups .

2) There is no inclusion/ exclusion criteria with regards to the molecule studied. As authors themselves have opined, there are non cardiac factors which can  affect the klotho molecule levels  which have not been mentioned.

3) The low cardiovascular risk profile of the group is a major limitation.

4) There is no mention of cardiac rehab modification of klotho levels translating into reduced coronary events. Hence the significance of this remains unclear.

I strongly feel you should do a proper scientific study find association between cardiac rehab and klotho levels in patients with cardiovascular disease.

Round 2

Reviewer 1 Report

Comments and Suggestions for Authors

I congratulate the authors for the improved version of the proposed manuscript. The authors have taken into account the suggestions given.

Reviewer 2 Report

Comments and Suggestions for Authors

Congrats for the revision

Reviewer 3 Report

Comments and Suggestions for Authors

Dear Authors,

It seems that the authors have made the changes requested. I think there is no obstacle to publishing the article in this form in the journal.